# Effect of Zhan Zhuang Qigong on upper limb static tremor and aerobic exercise capacity in patients with mild-to-moderate Parkinson's disease: study protocol for a randomised controlled trial

Linlin Zhang ![ORCID],[1] Xihua Liu,[2] Xiaoming Xi,[1] Yuxiao Chen,[1] Qing Wang,[1] Xinjie Qu,[1] Haihao Cao,[1] Limin Wang,[1] Wenyu Sun,[2] Guoming Chen,[2] Huifen Liu,[1] Xiaoyu Jiang,[1] Hang Su,[1] Jiahui Jiang,[1] Hongyan Bi[2]

[1]Shandong University of Traditional Chinese Medicine, Jinan, china
[2]Shandong University of Traditional Chinese Medicine Affiliated Hospital, Jinan, Shandong, China

**Correspondence to**
Professor Hongyan Bi;
Hy__bi@163.com

## ABSTRACT

**Introduction** Currently, the first choice for the clinical treatment of static tremor in Parkinson's disease (PD) is drug therapy, however side effects are common. In recent years, the effects of physical therapy on PD has become a serious research focus. Studies have indicated that aerobic and resistance exercises alleviate PD movement disorders and improve aerobic capacity, but the effects of Qigong on PD static tremor and aerobic capacity remain unknown.

Methods and analysis

**Objective** To observe the effects of Zhan Zhuang Qigong on upper limb static tremor and aerobic capacity in patients with PD, we established a rigorous randomised, parallel-controlled, assignment hidden, evaluator-blinded protocol.

**Methods** Seventy-two patients with PD, at the Affiliated Hospital of Shandong University of Traditional Chinese Medicine, were recruited and randomly divided into a control (n=36) and experimental group (n=36). The intervention group received conventional medicine plus Zhan Zhuang Qigong exercises five times a week at 30 min each time, over an 8-week period. The long-term effects of Zhan Zhuang Qigong on PD was investigated after the intervention. Phyphox APP, CRST, CPET, UPDRS(ll, lll) were used to evaluate tremor, aerobic capacity, and motor function in groups.

**Discussion** We are investigating the effects of Zhan Zhuang Qigong on upper limb static tremor and aerobic capacity in patients with PD. If positive are identified, they will add a new research direction and evidence for the clinical exploration of exercise therapy for PD.

**Ethics and dissemination** This study was approved by the Ethics Committee of Shandong University of Traditional Chinese Medicine (Approval Number: 2021–025-KY). The Committee will be informed of any changes to the trial protocol, such as intervention intensity, outcome indicators and data collection. Study results will be presented as a paper at an international conference or in a journal.

**Trial registration number** ChiCTR2100053529.

## STRENGTHS AND LIMITATIONS OF THIS STUDY

⇒ Currently, the clinical effects of physical therapy toward Parkinson's tremor are not ideal, with no clinical studies conducted on the effects of Qigong on Parkinson's tremor.
⇒ The mobile phone tremor test and cardiopulmonary exercise test used in our outcome assessments were clinically verified as accurate and reliable.
⇒ Our rigorous hidden allocation, randomised parallel controlled trial design is more reliable and accurate than pre-trial and post-trial controls previously used in Parkinson's disease clinical trials.
⇒ Staff turnover during the study intervention period may lead to bias in the final results.
⇒ We are unable to investigate other forms of Parkinson's tremor other than static tremor.

## INTRODUCTION

Parkinson's disease (PD) is a degenerative disease of the central nervous system common in middle-aged and elderly individual.[1] The PD prevalence in individuals aged 65–69 is approximately 0.5%–1%, and in individuals aged over 80 is 1%–3%. With increased population ageing, it is anticipated that PD prevalence and incidence will increase by more than 30% by 2030, and place tremendous pressure on families and society.[2]

As one of the main motor symptoms of PD, static tremor is characterised by rhythmic and clay ball-like tremor and is accompanied by a large amplitude and frequency of 4–6 Hz, which occurs more when the weight of affected limbs or other body structures is supported. The tremor is aggravated when general exercises (such as walking) or pressures increases, but is alleviated or disappears when the body moves at will. The tremor gradually spreads from the distal end

of one limb to the other.[3 4] Parkinson's tremor is one of the most troubling symptoms in patients with PD, which not only reduces daily living activities, but also imbues a sense of inferiority, depression and reduces participation in society.[5] Currently, the most effective clinical treatment for PD remains drug therapy.[2 6 7] Levodopa is considered the best treatment compound for supplementing exogenous dopamine and controlling PD, but its long-term use gradually reduces its efficacy and leads to central and peripheral adverse reactions.[8–10] Moreover, levodopa elicits poor responses to PD static tremor and has no effect on non-motor symptoms.[11] Invasive surgical treatments, such as focused thalamotomy and deep brain stimulation provide significant tremor relief in the short term, but do not prevent neurodegenerative progression in underlying PD.[8 12–14] Additionally, surgery is risky and may be accompanied by complications.[12]

In recent years, PD symptom improvement by exercise therapy has gained considerable research traction. Several studies reported that aerobic and resistance exercises alleviated PD motor symptoms, including tremor.[15–19] Among traditional Chinese Qigong therapies, Tai Chi, Wuqinxi and Baduanjin are believed to enhance balance in patients with PD and correct abnormal gait. However, few studies have focused on PD tremor and aerobic capacity.[20]

Zhan Zhuang Qigong is a basic technique in Tai Chi, and is a static, moderate and low intensity aerobic exercise combining body and mind.[21 22] Several clinical studies have reported the benefits of Tai Chi for PD motor and non-motor functions,[23] Tai Chi is a low cost exercise. It reduces falls, improves posture control, improves walking ability, improves depression and insomnia symptoms and improves quality of life. As a basic Tai Chi skill, the Zhan Zhuang has many Tai Chi characteristics.[24] For example, Zhan Zhuang movements combine body and mind, they control limb and trunk stability and change the centre of gravity with the mind. Additionally, like Tai Chi, Zhan Zhuang is also a closed circular motion technique which maintains action continuity and accuracy via adjustment, feedback, re-adjustment and re-feedback, which improve the body's recognition of stimuli and improves involuntary tremor.[25 26]

## Methods and analysis

The study is a two-arm, randomised, parallel-controlled trial with assignment-hiding and evaluator blindness. The main objective is to investigate the effects of Zhan Zhuang on upper limb static tremor and aerobic capacity in patients with mild-to-moderate PD, so as to provide a safe, effective, low-cost and easy-to-master aerobic exercise therapy for patients with PD.

## Patient and public involvement

The public is not involved in the study design. However, we will adjust intervention intensity and other details based on how participants respond to the intervention.

## Study design and setting

The study will be conducted in the Affiliated Hospital of Shandong University of Traditional Chinese Medicine. We will include 72 patients with PD with obvious upper limb static tremor. Recruitment will begin on 1 June 2022 and run until 1 June 2023. Using the random number table method, 72 patients will be randomly divided into an intervention and control group in a 1:1 ratio. Basic patient information, including gender, age, disease course, education level, residence status and drug-use will be collected. Tremor and motor function tests will be performed at least 8 hours after patients stop taking medication.

The control group will be treated with conventional medicine and routine aerobics. According to the Chinese Guidelines for the Treatment of PD (fourth edition), walking will be selected as the aerobic exercise mode in this group,[27] five times a week at 30 min each time for 8 weeks. A combined cardiopulmonary exercise test (CPET) (treadmill test) will be used to measure peak oxygen uptake (peakVO$_2$) in patients with PD at baseline and after intervention, and also anaerobic threshold (AT) and maximum heart rate (HRmax). During training, patients exercise intensity will be controlled at moderate exercise intensity (HRmax (60%–80%) measured by CPET or 11–13 Rating of Perceived Exertion (RPE). Changes in peakVO$_2$ and AT at baseline and after intervention will be evaluated for improvements in aerobic exercise capacity. Walking will be performed about 1 hour after patients take their medication, and the tremor symptoms were resolved or disappeared.

The intervention group will take conventional medicine combined with Zhan Zhuang Qigong. An experienced Qigong master will guide Zhan Zhuang exercise. Week 1 will comprise the learning phase, while weeks 2–8 will be the practice phases, to be performed five times a week at 30 min each time. Intensity HRmax (60%–80%) will be measured using a cardiopulmonary combined exercise test or RPE test grade 11–13. Patients will be issued a training instruction manual and a personal exercise calendar to record the number of weekly exercises and time of each exercise. Zhan Zhuang training will be about 1 hour after patients take their medication, and tremor symptoms were relieved or disappeared. One auxiliary staff member will oversee safety measures.

## The main points of Qigong movement of Zhan Zhuang

Individuals should stand naturally, with feet about 30 cm apart and parallel to the front; arms around, with naturally open hands and two fingertips about 15 cm apart; flex the knee hip about 5°–10° and tilt the hip forward (figures 1 and 2).

## Participants and recruitment

Using the Affiliated Hospital of Shandong University of Traditional Chinese Medicine as the base, we will visit and recruit patients from the rehabilitation and neurology

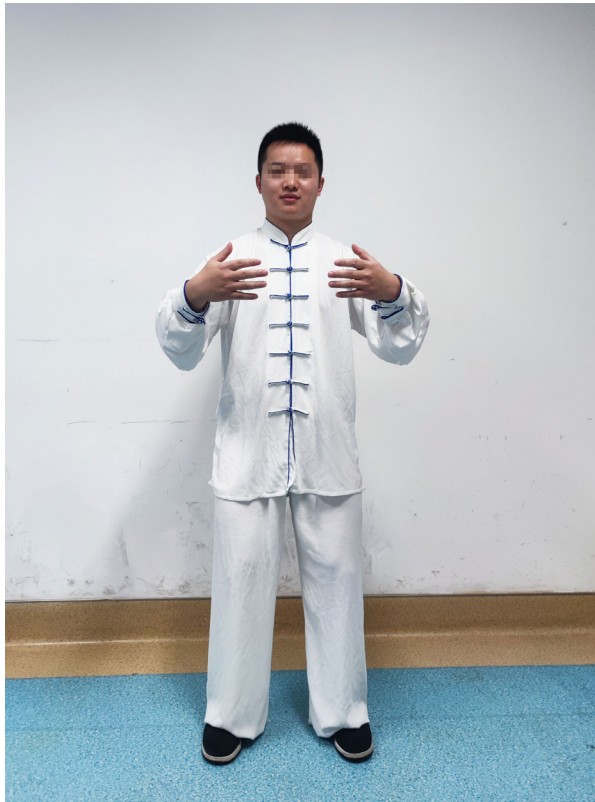

**Figure 1** Demonstration of Qigong movement of Zhan Zhuang.

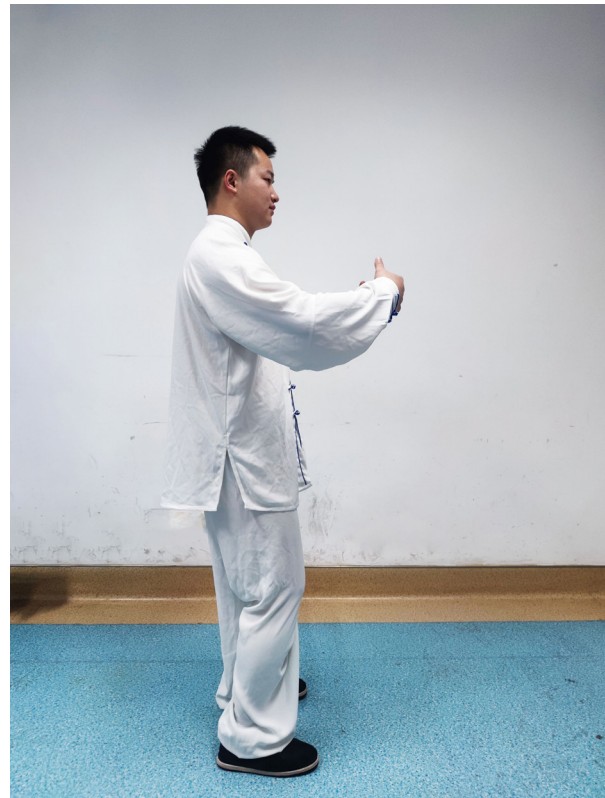

**Figure 2** Demonstration of Qigong movement of Zhan Zhuang.

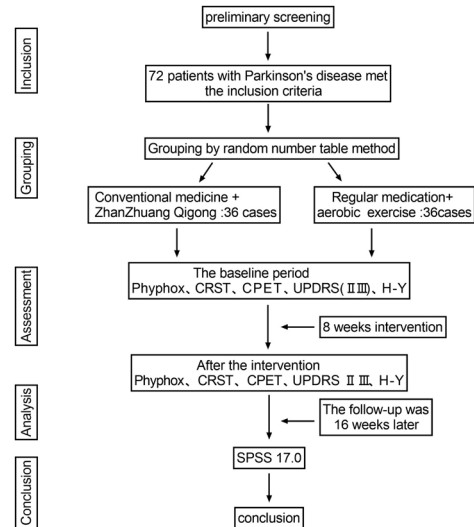

**Figure 3** Test flow chart. CPET, cardiopulmonary exercise test; CRST, Clinical Tremor Rating Scale; UPDRS, Unified Parkinson's Disease Rating Scale.

departments, and post recruitment advertisements on WeChat.

We will assess upper limb static tremor according to the Unified Parkinson's Disease Rating Scale (UPDRS) Part III and the Hoehn-Yahr classification (stages 1–3) (see figure 3).

### Inclusion criteria

1. Primary PD agrees with recommended Chinese Clinical Diagnostic Criteria for PD (2016).
2. 50–70 years old.
3. Improved Hoehn-Yahr stage 1–3.
4. Stable condition, taking Madopar and standing for the 'prescribed period'.
5. No serious cognitive dysfunction, has the ability to learn Zhan Zhuang movements; (Mini-Mental State Examination score, illiteracy >17, primary school >20 and middle school >22).
6. Agrees to participate, and sign an informed consent sheet.

### Exclusion criteria

1. Secondary Parkinson's syndrome and Parkinson's superposition syndrome caused by cerebrovascular disease, drugs and other causes.
2. Patients with serious heart, lung, liver, kidney dysfunction or cancer.
3. Those individuals who cannot stand due to serious knee diseases, for example, severe knee osteoarthritis and knee trauma.
4. Patients with dementia or serious mental illness.
5. Patients unable to stand or walk independently due to fracture, surgery or other reasons.
6. No tremors.
7. The 'opening period' tremor is obvious after taking medication.

### Termination/shedding criteria

1. Patients not cooperating with prescribed exercises or exercising poor compliance.
2. Patients whose condition deteriorates or complications arise during the study should not continue.

### Sample size

This study is designed as a randomised controlled trial. The main outcome is the mean relief of upper limb static tremor after treatment. A review of previous studies showed that aerobic exercise could reduce upper limb static tremor amplitude by approximately 60%. It is estimated that the average tremor amplitude of the control group is approximately 76±136 mm, and the intervention group is expected to be reduced by 50 mm after training. The test level α is set at 0.05 (bilateral), the degree of certainty 1-β is set at 0.80 and the control:intervention group ratio=1:1. A sample size of 63 was calculated using *http://powerandsamplesize.com.* However, considering an abscission rate of about 15%, 72 patients with PD will be included.

### Randomisation and allocation concealment and blinding

This study is single-blinded. Because of the Qigong intervention, training personnel and subjects are not blinded, however testers, data collectors, data scientists and analysts are blinded. Implementation of training, evaluation and statistical analysis of the separation of the way, that is, the training personnel do not contact the test personnel. Before testing, the intervention and control groups were renamed as groups A and B, respectively. The test tester is only responsible for the test by number, and does not tell the group of the test subject. Data collection and processing personnel are separated; data collection personnel obtain group A or group B date based on the number and prove this information to the data statistician for statistical analysis. To ensure study objectivity, quality control personnel will identify groups after statistics are completed.

## ASSESSMENT RESULT
### Primary outcome

(1) Static tremor of the upper limbs

①  Mobile phone movement exercise test

Phyphox APP[28] is based on the Android system (V.1. 1. 9). It collects the static tremor date of the upper limb of patients with an obvious tremor, at a frequency of 100 Hz over 30 s.

Data collection on the static tremor: The patient relaxes, sits in a chair, places his/her hands on the armrest and holds the mobile phone with the screen up on the shaking side. At the same time, patients perform numerical subtraction calculations to distract their attention. The time from the beginning of the tremor is recorded over 30 s.

②  The Clinical Tremor Rating Scale (CRST) A+B, CRST-A is used to evaluate the tremor at rest, postural and intentional; CRST-B assesses handwriting (dominant hand only), wide and narrow spiral drawing, straight lines and casting. For the dominant hand, the maximum subscore of hand tremor is 32 for CRST (A+B) and 28 for the non-dominant hand. The higher the score, the more severe the tremor.

(2) Aerobic capacity

Combined Cardiopulmonary Exercise Test (CPET)

The cardiovit CS-200 cardiopulmonary combined exercise test system (SCHILLER, Switzerland) is used for CPET. The treadmill test is performed, and the increasing rate set at 10 W/min. Based on standards from Harbor-UCLA Center Cardiopulmonary Exercise Laboratory, USA, the resting state lung function examination, including fast lung capacity, slow lung capacity, maximum minute ventilation capacity and diffusion function, should be completed first. Symptom limiting maximum CPET is then completed using continuous ramping regimen.[29–31] CPET is divided into four stages, including 3 min of rest, 3 min warm-up (no treadmill power), then exercise at an increasing power rate of 10 W/min until symptom limiting maximum exercise, followed by more than 5 min of recovery.[32] Continuous 12-lead ECG, non-invasive cuff blood pressure and oxygen saturation measurements are required during the CPET process. PeakVO$_2$, AT, and HRmax are measured and recorded.

### Secondary indicator

Motor symptoms and daily living abilities

1. UPDRS II and III: The scale is divided into four parts. The first part evaluates the patient's mental behaviours and emotions. Part II assesses the patient's ability to live a daily life. Part III assesses the patient's motor functions. The total score in part IV reflects patient complications. We selected Parts II and III to evaluate the daily living activities and motor functions of patients, respectively.
2. Revised Hoehn-Yahr Grading Scale: The scale is divided into eight stages, from 0 to 5 (0, 1, 1.5, 2, 2.5, 3, 4 and 5), to evaluate disease severity.

### Adverse events

Adverse events (AEs) occurring during the study are recorded using a Case Report Form. For this trial, an AE will be defined as adverse and unexpected signs, symptoms or diseases associated with the intervention, such as falls, joint injuries, hypertension, etc. Participants will be asked about their medical history and any AEs they have experienced before training begins, and AEs will be recorded after each intervention. The principal investigator will be notified within 24 hours if a serious AE occurs. If subjects withdraw from the trial due to any AE, the study team will follow-up accordingly or until the AE is resolved, This information will be recorded in the original file. The principal investigator manages all safety reports. Any AEs relating to the intervention will be reported to the Ethics Review Committee of the Affiliated Hospital of Shandong University of Chinese Medicine.

**Table 1** Data collection schedule and visit assessments

| Measures | Baseline week | Intervention period (0–8 weeks) | End of intervention (9 weeks) | Follow-up period (9–16 weeks) | Follow-up (17 weeks) |
|---|---|---|---|---|---|
| Participant characteristics | √ | | | | |
| Mobile phone tremor test | √ | | √ | | √ |
| CRST (A+B) | √ | | √ | | √ |
| CPET | √ | | √ | | √ |
| UPDRS (II, III) | √ | | √ | | √ |
| Hoehn-YaHr | √ | | √ | | √ |
| Adverse events[*] | | √ | | √ | |
| Combined medication[†] | | √ | | √ | |

*Adverse events: Adverse event at any visit during treatment sessions and over the 36 weeks will be monitored. The research team will review all trial protocols, monitor patient safety and investigate any adverse events.
†Combined medication: Patients will be asked if they have used other medications during the treatment. If they have, medications type and dose will be recorded.
CPET, cardiopulmonary exercise test; CRST, Clinical Tremor Rating Scale; UPDRS, Unified Parkinson's Disease Rating Scale.

## Statistical analysis

We will evaluate the outcome and collect data at week 9 of the intervention and follow-up at week 17 of the intervention (table 1). Basic patient information, mobile phone exercise test data, CRST scores and UPDRS (II and III) scores will be exported to our computer system CSV format.

SPSS V.17.0 statistical software will be used for data processing and analysis: (1) Measurement data will be processed using the mean SD ($x^2 \pm s$); (2) T-tests will be used to assess: normal distribution and homogeneity of the overall variance in both samples; (3) T-tests will be used to assess: normal distributions, but uneven overall variance; (4) The Wilcoxon rank-sum test will assess if date does not conform to a normal distribution; p value<0.05 indicates statistical significance.

## Mobile exercise test data

The original form is the acceleration a (m/s) of three axes (X, Y and Z) vector direction, respectively, rotation radian w (rad/s). The MATLAB (version R 2018a, MathWorks, USA) computer program will be used to perform Fourier transformation of the data, and acceleration and rotation radian measurements by smartphone will be converted to corresponding frequency, peak amplitude and speed date. Final phone test data will be presented as the amplitude and frequency of static tremors. Then, normal distribution tests will be conducted. Since the sample size is small (N<5000), the Shapiro-Wilk normal test will be used. P value>0.05 indicates a normal distribution, while p value<0.05 indicates a non-normal distribution.

According to normal distribution test date, if mobile phone exercise test data are normally distributed and the variance uniform, mobile phone exercise test data in patients with PD in different groups will be tested using with two sample independent t-tests. P value<0.05 will indicate if the exercise is statistically significant between groups. If the data are not normally distributed, non-parametric tests (median and Mann-Whitney U tests) will be performed for comparison. P value<0.05 in median tests will indicate if differences in central tendency scores between groups are statistically significant.

## CPET test data

After subjects complete the CPET test, raw date will be saved. According to standardised principles of CPET data analysis, and the interpretation of Food and Drug Administration clinical trials in the Harbor-UCLA Medical Center,[33 34] original breath-by-breath data from subjects will be sensed every second from the CPET system software. Then, average the data for 10s and export the average data.[31 35] The warm-up value is the average of the last 30s of the warm-up period, the anaerobic threshold value is the data value of 10s and the peak value is the average of 30s during the extreme exercise. Anaerobic threshold analysis is performed using a v-slope method, that is, the mean value of 10s of oxygen uptake when the $CO_2$ emission rate is significantly accelerated relative to oxygen uptake rate during movement.[36]

## DISCUSSION

PD not only reduces patients quality of life, but also generates a heavy economic burden for society and families. Upper limb static tremor is one of the main PD motor symptoms. Dyskinesia can also lead to decreased activity and aerobic capacity in patients with PD. Currently, non-drug therapy for PD mainly includes aerobic exercise and resistance training. Previous studies reported those methods were beneficial for tremor, bradykinesia, balance and quality of life in patients with PD.[37] However, the effect of Qigong on PD static tremor and aerobic ability remains unknown.

Free radicals (OH)[38] are a strong oxides that induce lipid peroxidation in unsaturated fatty acids and produce lipid peroxidation, which damages proteins and DNA

and lead to cell degeneration and death. Oxidative stress[39] represents an imbalanced state between oxidation and antioxidant effects in the body, which are important factors contributing to ageing and disease. Due to increased activity of B-monoamine oxidase in patients with PD, excessive free radicals oxidise and destroy cell membranes.[40] At the same time, dopamine oxidation products in substantia nigra cells polymerise to form neuromelanin, which combines with iron to produce Fenton reactions and free radicals. Under normal circumstances, antioxidant molecules in brain cells are plentiful and include, glutathione (GSH), glutathione peroxidase (GSH-PX) and superoxide dismutase (SOD), thus free radicals generated by dopamine oxidation will not lead to oxidative stress.[41 42] In PD, the substantia nigra is vulnerable to oxidative stress due to decreased ferritin and glutathione levels, and increased lipid peroxide, and iron ion concentrations.[41–43] Previous studies[11 44] reported that Zhan Zhuang improved SOD the activity levels in the body, prevented harm from superoxide free radicals and exerted positive roles fighting ageing, inflammation and immune disease, Therefore, the Zhan Zhuang may protect the substantia nigra in patients with PD by increasing SOD activity levels, reducing substantia nigra lesions, slowing down PD progression and improving overall symptoms in patients.

Currently, the generally accepted PD tremor mechanism in clinical practice is the induced-switch-regulation model proposed by Duval.[4]

Induced: Pathological activity in the basal nucleus striatum induces rhythmic cluster discharge activity in thalamic neurons via the excessive input of inhibitory signals to thalamic cells, and changes the discharge mode.

Switch: Oscillations in internal and external networks of the thalamus combine and cause oscillations to continuously integrate and strengthen until they reach tremor production threshold values.

Regulation: After PD tremor-related oscillations are introduced to the cerebellum, it processes oscillation as random movements and adjusts tremors to ensure stability in oscillation activities, such as frequency and amplitude. Tremors transmitted to the cerebral cortex are then transmitted to the periphery via the corticalspinal cord pathway in the form of oscillations, and are finally manifested body tremors. Yang *et al*,[45] based on data from basal ganglia injury mice and rats models, reported that exercise training reduced over-activation of basal ganglia cortical circuits, thereby weakening tremor signal transmission, and alleviating tremors. Recent studies also reported these animal models were applicable to humans.[46] Kadkhodaie *et al*[15] observed that upper limb centrifugal exercises effectively reduced upper limb tremor amplitude in patients with PD. Zhan Zhuang as an aerobic exercise may also reduce tremors at the motor level.

PD tremor is influenced by attention and stress levels.[47] Previous studies reported that tremors were intentionally attenuated when attention was focused on them, and that

PD tremors were is amplified by stress.[48] Wang[49] using nuclear magnetic resonance to analyse tremor subtypes in patients with PD, observed that the brains of tremor subtypes in patients with a wide range of functional connection between the frontal lobe and cerebellum, and UPDRS score III movement and the cognitive function score also existed significant correlation, that movement function and cognitive function in patients with PD has a high degree of correlation. DA substitution therapy also alleviated some motor characteristics in PD, but the observed beneficial effects on cognitive function were small.[50] Several studies[21 51 52] reported that Zhan Zhuang enhanced the attention and cognitive ability of adolescents and the elderly. Yiyue Chen[53] observed that Zhan Zhuang exercises increased P3 wave amplitude in frontal, central and top regions, with a shorter incubation period when compared with baseline. P3 wave amplitude reflected the allocation of neural electrical resources, and latency which was related to the speed of stimulus classification. These results suggest that station post affected neural electrical processing related to cognitive control, by changing neural electrical resource allocation and stimulus classification speeds, thereby affecting cognitive function. Therefore, the station post may indirectly improve tremor performance by enhancing cognitive ability in patients.

Executive control is a general control mechanism where individuals coordinate different cognitive processes and resources to ensure their cognitive system implements specific goals in a flexible and optimised way when completing complex cognitive tasks. Executive control also coordinates brain commands and muscle movements. Yang *et al*[54] conducted a 1-year Zhan Zhuang exercise in healthy college students, and observed that EEG activity coefficients in students who participated in Zhan Zhuang exercises were generally decreased, and that EEGs showed a high synchronisation trend. Whole brain ordering was improved, and brain activities were more coordinated and synchronised. Fang[55] reported that Zhan Zhuang exercised the muscles of the whole body statically, provided benign stimulation to the brain and adjusted the disorder of excitement and inhibition of the cerebral cortex. Jianguo[56] also confirmed that Zhan Zhuang improved subject response speeds and accuracy, and also improved the executive control levels in the brain. Zhan Zhuang combines body and mind, it focuses on controlling the body with the mind, and mobilising muscle groups in the trunk and limbs to participate in this process.[21] This body and mind combination stimulates the nervous system and increase its sensitivity, ensuring more efficient and precise muscle control. Additionally, Zhan Zhuang places the body in a stressed state of postural control for long periods, with previous clinical studies showing that stress significantly affects neural plasticity.[56] Thus, higher levels of executive control and finer, more efficient postural control may benefit PD tremor relief.

Guo[21] observed that B cell receptor and NF-κB signalling pathways were significantly enriched in college

students after Zhan Zhuang, suggesting the technique may enhance in vivo immunity and improve inflammatory microenvironments by activating B cell receptor signalling and improving T lymphocyte levels, or regulating NF-κB signalling. Regulating valine, leucine and isoleucine metabolism, prevents excessive glutamate accumulation, increases astrocyte and 5-hydroxytryptamine levels, and prevents neuronal necrosis and apoptosis. The specific regulation of the lysosomal pathway mediates autophagy levels, improves neuron survival rates and alleviates depression and improves adverse psychiatric symptoms. Several studies[21 56–61] reported that the station post significantly improved blood flow, improved the inflammatory environment and possibly enhanced immunity.

### Ethics and dissemination

This study has been approved by the Ethics Committee of Shandong University of Traditional Chinese Medicine (Approval Number: 2021–025-KY). The results of this study will be published in a peer-reviewed journal, and trial participants will be informed via email or phone calls.

**Contributors** All authors contributed to the study design. HB and XL conceived the research idea. LZ and XX designed experiments. YC, QW, XQ, HC and LW steered the ethics application, and GC and WS contributed to data analysis. JJ, HS, XJ and HL drafted the manuscript. All authors approved the final version of the manuscript.

**Funding** This study was funded by the Science and Technology Project of Traditional Chinese Medicine of Shandong Province sponsored by Shandong Provincial Health Commission (Grant Number: 2020Z04). However, the sponsor was not involved in the conduct of the trial, date collection and statistical processing. Final trial results and the publication of any reports are independent of the sponsor.

**Competing interests** None declared.

**Patient and public involvement** Patients and/or the public were involved in the design, or conduct, or reporting, or dissemination plans of this research. Refer to the Methods section for further details.

**Patient consent for publication** Consent obtained directly from patient(s).

**Provenance and peer review** Not commissioned; externally peer reviewed.

**ORCID iD**
Linlin Zhang http://orcid.org/0000-0002-7692-087X

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
