## [Reviewer comments · BMJ Open]

ARTICLE DETAILS

TITLE (PROVISIONAL)	Effect of ZhanZhuang Qigong on upper limb static tremor and aerobic exercise capacity in patients with mild to moderate Parkinson's disease: Study Protocol for a Randomized controlled Trial
AUTHORS	zhang, linlin; Liu, Xihua; Xi, Xiaoming; Chen, Yuxiao; Wang, Qing; Qu, Xinjie; Cao, Haihao; Wang, Limin; Sun, Wenyu; Chen, Guoming; Liu, Huifen; Jiang, Xiaoyu; Su, Hang; Jiang, Jiahui; Bi, Hongyan

VERSION 1 – REVIEW

REVIEWER	Catarina Godinho Centro de Investigação Interdisciplinar Egas Moniz
REVIEW RETURNED	22-Dec-2021

GENERAL COMMENTS	This is a very interesting study protocol and I only have a few comments to help to improve the manuscript. Comment 1: The references to this sentence: A number of studies have shown that aerobic exercise and resistance exercise are beneficial to the alleviation of PD motor symptoms including tremor 6 14-19. Should be reassessed. Given there are no studies objectively testing exercise and tremor; all these references insinuating results over tremor should be clearer. Comment 2: However, As Tai Chi is a highly difficult sport, most PD patients are middle-aged and elderly, and their learning ability is reduced. In addition, due to the limitations of venue and practice frequency, it takes at least 8 weeks to master it from the beginning to the preliminary stage, and even longer to master it well. This is an integration rationale, given the general idea is that tai chi is low impact ready to follow modality of training. It is controversial and maybe not clear if we are using the same definition of what tai chi is in PD. Better to build on the positive effects of Tai chi and say this adds on by....than to exaggerate the limitations of tai chi per se. Comment 3: Reference to the impact of tremor on patients day to day could better enrich the introduction and discussion. The discussion is too dense and complex. English should be reviewed and simplified. Comment 4:
--

	In Page 6 line 10 “The average value of tremor was estimated to be (76±136)” ... the unit of measurement is missing and why are the values presented in parentheses? Comment 5: Please rectify all the typos writing in manuscript
--	--

REVIEWER	Yanjie Zhang The Chinese University of Hong Kong - Shenzhen, Physical Education Unit
REVIEW RETURNED	15-Jan-2022

GENERAL COMMENTS	The author reported on Effect of ZhanZhuang Qigong on upper limb static tremor and aerobic exercise capacity in patients with mild to moderate Parkinson's disease: Study Protocol for a Randomized controlled trial. Overall, there are numerous places in the text where there are errors, poorly constructed sentences, and even places where the text is incomprehensible. I recommend the authors seek the assistance of a native English-speaker. And it is necessary to revise the following points.  1. Please use “aerobic capacity” instead of “aerobic exercise ability” throughout the whole content. 2. Line 10, please delete the space between 0.5-1%. 3. Line 18-20, It is aggravated when general exercise or pressure increases, and alleviates or disappears when the body moves at will. And gradually spread from the distal end of one limb to the other. What's meaning of general exercise? Could you give some examples for explanation? In general, this sentence is difficult to understand. Please rewrite it. 4. Page 3, Line 41-42. ZhanZhuang Qigong is the basic technique of Tai Chi, and it is a static, medium and low intensity aerobic exercise combining body and mind. Please change the “medium” to “moderate”. 5. Line 45, please unify the spelling of the word. Such as “As”. 6. Page 4, Line 41-42, Tests for tremor and motor function were performed at least eight hours after patients stopped taking the drug. Why do you choose 8h as the cut-off time for measurements? 7. Page 4, Line 50, “Patients' exercise intensity reached HRmax (60%~80%) or self-induced fatigue Scale (RPE) grade 11~13.” You mean the exercise intensity is moderate intensity exercise with HRmax (60%-80%) or RPE at 11-13. The sentence is not well expressed. Please rewrite this sentence. 8. Page5, Line 1, “Invite an experienced qigong master to guide the ZhanZhuang”. Please rewrite. 9. Page5, Line 1, “The intensity reached HRmax (60%~80%) or selfinduced fatigue Scale (RPE) grade 11~13”. In your study, how do you make sure that the intensity of exercise is within that range during practice Zhanzhuang. 10. Page 7, Line 20-21. Blind test testers, data collectors, data statistics and analysis evaluators. Please rewrite. 11. In the discussion part, you cited a large number of literature studies, some of which were too detailed, so you can mainly explain the research results of your study in future manuscript.
--

VERSION 1 – AUTHOR RESPONSE

Reviewer: 1

Reply to Professor Catarina Godinho

Thank you very much for your recognition and suggestions, and we have modified the manuscript according to your suggestions.

Comment 1:

The references to this sentence: A number of studies have shown that aerobic exercise and resistance exercise are beneficial to the alleviation of PD motor symptoms including tremor 6 14-19. Should be reassessed. Given there are no studies objectively testing exercise and tremor; all these references insinuating results over tremor should be clearer.

-Response:

I am so sorry for the inappropriate references, which are really not objective enough. So we switched to more objective references to clinical trials and Meta-analysis.

Comment 2:

However, As Tai Chi is a highly difficult sport, most PD patients are middle-aged and elderly, and their learning ability is reduced. In addition, due to the limitations of venue and practice frequency, it takes at least 8 weeks to master it from the beginning to the preliminary stage, and even longer to master it well.

This is an integration rationale, given the general idea is that tai chi is low impact ready to follow modality of training. It is controversial and maybe not clear if we are using the same definition of what tai chi is in PD. Better to build on the positive effects of Tai chi and say this adds on by....than to exaggerate the limitations of tai chi per se.

-Response:

The integration rationale is indeed controversial, so we modified it into the positive effect of Tai Chi in Parkinson's disease according to your suggestion.

Comment 3:

Reference to the impact of tremor on patients day to day could better enrich the introduction and discussion. The discussion is too dense and complex.

English should be reviewed and simplified.

The introduction section adds to the adverse effects of Parkinson's tremor on life.

The discussion section has been simplified and the English has been proofread

-Response:

We add in the introduction to the adverse life effects of Parkinson's tremor. And the discussion section was simplified and the English was proofread.

Comment 4:

In Page 6 line 10 "The average value of tremor was estimated to be (76±136)" ... the unit of measurement is missing and why are the values presented in parentheses?

-Response:

I'm so sorry, this is a clerical error. The value is the tremor magnitude. We modified and added units (millimeters).

Comment 5:

Please rectify all the typos writing in manuscript

-Response:

I'm very sorry that many language errors in the manuscript have affected understanding and reading. We consulted native English speakers to proofread and revise the manuscript.

Reviewer: 2

Reply to Dr. Yanjie Zhang

Thank you very much, Dr. Zhang, for your suggestions. I apologize for the difficulty of reading and understanding the manuscript due to English problems. We carefully revised the manuscript according to your suggestions, and consulted experts who native English-speaker to polish the English of the manuscript.

1. Please use “aerobic capacity” instead of “aerobic exercise ability” throughout the whole content.

-Response:

We are sorry for the word error and have corrected it according to your suggestions.

2. Line 10, please delete the space between 0.5-1%.

-Response:

We deleted the space between 0.5-1%.

3. Line 18-20, It is aggravated when general exercise or pressure increases, and alleviates or disappears when the body moves at will. And gradually spread from the distal end of one limb to the other. What's meaning of general exercise? Could you give some examples for explanation? In general, this sentence is difficult to understand. Please rewrite it.

-Response:

I am sorry that this sentence is not clear enough. We reedited the sentence to clarify our message. General exercises refer to exercises that can be mastered without special professional training and are often done in daily life. Such as walking, jogging and so on.

Rewrite : The tremor is aggravated when general exercises (such as walking) or pressures increases, but is alleviated or disappears when the body moves at will.

4. Page 3, Line 41-42. ZhanZhuang Qigong is the basic technique of Tai Chi, and it is a static, medium and low intensity aerobic exercise combining body and mind. Please change the “medium” to “moderate”.

-Response:

We corrected the word mistakes.

5. Line 45, please unify the spelling of the word. Such as “As”.

-Response:

We unified writing.

6. Page 4, Line 41-42, Tests for tremor and motor function were performed at least eight hours after patients stopped taking the drug. Why do you choose 8h as the cut-off time for measurements?

-Response:

The assessment will be chosen 8 hours after drug withdrawal to ensure that patients is in "OFF" period that the drug was less or less effective and abnormalities in motor function could be observed. Second, the same patient's motor function at baseline and after the intervention will be assessed strictly at the same discontinuation time, both to rule out drug influence on the outcome.

7. Page 4, Line 50, “Patients' exercise intensity reached HRmax (60%~80%) or self-induced fatigue Scale (RPE) grade 11~13.” You mean the exercise intensity is moderate intensity exercise with HRmax (60%-80%) or RPE at 11-13. The sentence is not well expressed. Please rewrite this sentence.

-Response:

I'm sorry this sentence is not clear. We have redefined it.

Rewrite: During training, patients exercise intensity will be controlled at moderate exercise intensity (HRmax (60%-80%) measured by CPET or 11-13 self-induced Fatigue Scale (RPE)) .

8. Page 5, Line 1, “Invite an experienced qigong master to guide the ZhanZhuang”. Please rewrite.

-Response:

We rewrote the sentence.

Rewrite: An experienced Qigong master will guide ZhanZhuang exercise.

9. Page5, Line 1, “The intensity reached HRmax (60%~80%) or self-induced fatigue Scale (RPE) grade 11~13”. In your study, how do you make sure that the intensity of exercise is within that range during practice Zhanzhuang.

-Response:

Zhanzhaung can increase the exercise intensity by changing the bending Angle of the knee joint. The greater the bending Angle of the knee joint is, the stronger the intensity is. During zhanzhaung intervention, the Angle of knee flexion will be gradually increased and a portable wearable smart device (such as an Apple Watch) is used to monitor heart rate or determine whether the required intensity is achieved based on the patient's self-fatigue level.

10. Page 7, Line 20-21. Blind test testers, data collectors, data statistics and analysis evaluators. Please rewrite.

-Response:

Rewrite:

This study is single-blinded. Because of the Qigong intervention, training personnel and subjects are not blinded, however testers, data collectors, data scientists, and analysts are blinded.

11. In the discussion part, you cited a large number of literature studies, some of which were too detailed, so you can mainly explain the research results of your study in future manuscript.

-Response:

Thank you very much for your reminding and suggestions. We simplified the discussion and corrected the language. We will explain the results in a future manuscript

VERSION 2 – REVIEW

REVIEWER	Catarina Godinho Centro de Investigação Interdisciplinar Egas Moniz
REVIEW RETURNED	10-Mar-2022

GENERAL COMMENTS	The authors responded appropriately to the reviewers' comments and suggestions. The study has a significant scientific interest and addresses a pertinent topic for which it should be published.
---

REVIEWER	Yanjie Zhang The Chinese University of Hong Kong - Shenzhen, Physical Education Unit
REVIEW RETURNED	27-Apr-2022

GENERAL COMMENTS	1. Page 3. Line59. "(HRmax (60%-80%) measured by CPET or 11-13 self-induced Fatigue Scale (RPE))". In the field of exercise, although the self-induced fatigue scale is commonly used to measure the physical activity level, it includes many different methods (e.g., RPE). In your study, the correct expression should be Rating of Perceived Exertion (RPE). 2. Please use a uniform front/space throughout the text. such as, in the discussion part, the word front and line space are different.
---

VERSION 2 – AUTHOR RESPONSE

Reviewer: 1

Reply to Professor Catarina Godinho

-The authors responded appropriately to the reviewers' comments and suggestions. The study has a significant scientific interest and addresses a pertinent topic for which it should be published.

- **Response:** Thank you very much for your previous revision suggestions, and thank you for your affirmation and support of the protocol.

Reviewer: 2

Reply to Dr. Yanjie Zhang

Thank you very much, Dr. Zhang, for your suggestions. I am very sorry that my English writing and expression skills are very poor, which may cause you trouble. I have corrected this error according to your advice.

1. Page 3. Line59. "(HRmax (60%-80%) measured by CPET or 11-13 self-induced Fatigue Scale (RPE))". In the field of exercise, although the self-induced fatigue scale is commonly used to measure the physical activity level, it includes many different methods (e.g., RPE). In your study, the correct expression should be Rating of Perceived Exertion (RPE).

-Response:

We have modified it according to your suggestion and replaced self-induced Fatigue Scale (RPE) with Rating of Perceived Exertion (RPE)

2. Please use a uniform front/space throughout the text. such as, in the discussion part, the word front and line space are different.

-Response:

We checked the manuscript and corrected the font, blank space and other formatting problems